# Profiling zero-dose measles-rubella children in Zambia: Insights from the 2024 post-campaign coverage survey

Moses Mwale[1]*, Guissimon Phiri[2], Francis Dien Mwansa[3], Peter J. Chipimo[1], Penelope Masumbu[1], Kennedy Matanda[2], Princess Kayeye[2], Kelvin Mwangilwa[4], Chola Nakazwe[5], Harriet Namukoko[5], Simon Mutembo[6], Freddie Masaninga[1], Jacob Sakala[2], Peter Clement Lugala[1]

1 World Health Organization, Lusaka, Zambia, 2 Expanded Program on Immunization, Child Health Unit, The Ministry of Health, Lusaka, Zambia, 3 UNICEF, Lusaka, Zambia, 4 Strategic Planning Information Management, Zambia National Public Health Institute, Lusaka, Zambia, 5 Population and Demography Unit, The Zambia Statistics Agency, Lusaka, Zambia, 6 International Vaccine Access Centre, Johns Hopkins Bloomberg School of Public Health, Baltimore, Maryland, United States of America

* mwalem@who.int

## Abstract

Measles & Rubella (MR) zero-dose children (unvaccinated for measles–rubella) cluster in underserved communities can sustain measles transmission. We estimated MR zero-dose prevalence after Zambia's 2024 MR Supplementary Immunisation Activity (SIA) and identified associated risk factors and barriers. A coverage survey (two-stage stratified cluster design) across all 10 provinces, was conducted from 27th December 2024–16th January 2025, involved interviewing caregivers of children aged 9–59 months; vaccination status was verified by card (11.7%) or recall (88.3%). Data were analysed using survey-weighted methods and logistic regression, adjusting for stratification, clustering, and sampling weights. Among 8,634 children, MR zero-dose prevalence was 11.97% (95% CI: 11.03–12.91), highest in Central (19.15%) and Western (17.71%), lowest in Copperbelt (6.69%). Urban residence reduced odds by 24% vs. rural (aOR 0.76, 95% CI: 0.63–0.92). Risks rose with age (>36 months: aOR 1.60, 95% CI: 1.27–2.00), maternal absence (aOR 1.74, 95% CI: 1.33–2.27), or death (aOR 2.40, 95% CI: 1.23–4.68). Most zero-dose children (88.75%) lacked other vaccines, indicating systemic gaps. Key barriers included unawareness (42.58%) and travel time (>2 hours: aOR 3.20, 95% CI: 1.43–7.16). Nearly one in eight Zambian children remained MR zero-dose post-2024 SIA, concentrated in rural, high-prevalence areas, older children, and motherless households. Priorities include health worker-led awareness campaigns, mobile services to cut travel time, and integrated SIA-Routine Immunisation (RI) strategies (microplanning, tracing, catch-up) to address systemic gaps, supporting global measles elimination under Immunisation Agenda 2030.

**Data availability statement:** The individual-level dataset analysed in this study is owned by the Zambia Ministry of Health and was accessed under a programme-related data-use agreement that prohibits public sharing of individual-level records. Researchers wishing to access the full dataset may submit a formal request in line with national data-governance procedures. Data may be requested from: Zambia Statistics Agency (ZamStats) – Office of the Statistician General Email: info@zamstats.gov.zm Ministry of Health – Expanded Programme on Immunisation (EPI) Contact: Dr Jacob Sakala, EPI Manager Email: sakalajac57@gmail.com.

**Funding:** The authors received no specific funding for this work.

**Competing interests:** The authors have declared that no competing interests exist.

## Introduction

Measles remains one of the most contagious vaccine-preventable diseases and continues to cause substantial morbidity and mortality worldwide. In 2022 alone, measles caused approximately 128,000 deaths globally, the vast majority among unvaccinated children in low- and middle-income countries (LMICs) [1]. Despite progress, 14.3 million children globally were classified as "zero-dose" in 2024—that is, they had not received any dose of a routine vaccine. This group remains a major driver of measles transmission through clustering in under-immunised communities [2,3]. Outbreaks in both LMICs and high-income countries underscore the continued threat posed by declining or stagnant immunisation coverage [4].

Zambia is no exception to these challenges. Between 2021 and 2024, the number of measles zero-dose children in Zambia rose from approximately 39,000–41,000, ranking the country among the top 12 in the Eastern and Southern Africa Region (ESAR) [5]. Nationally, measles elimination requires sustained ≥95% coverage with two doses of the measles-rubella (MR) vaccine, as outlined in the World Health Organization's (WHO) *Immunization Agenda 2030 (IA2030)* [6]. However, routine immunisation coverage has stagnated, and inequities persist, especially in rural and socioeconomically disadvantaged communities. These gaps threaten Zambia's ability to achieve measles elimination goals and mirror global concerns about stagnating immunisation progress [4,7].

Several studies have identified risk factors for zero-dose status, including rural residence, maternal absence or orphanhood, caregiver education level, and distance to health facilities [8–10]. However, many analyses rely on secondary datasets, lack disaggregated campaign-specific information, or fail to capture behavioural and social determinants of vaccine uptake. This limits the ability of policymakers to design context-specific interventions to close immunisation gaps. Locally generated evidence is especially critical as both Gavi's equity agenda and IA2030 prioritise reaching zero-dose children as a global public health imperative [6,11].

In Zambia, the 2024 Measles–Rubella Supplementary Immunisation Activity (SIA) was reported to have reached 97% coverage administratively, vaccinating 165,000 zero-dose children through targeted microplanning, community engagement, and digital monitoring [12]. Yet, administrative coverage often overestimates true performance. Independent evaluation is necessary to validate coverage, quantify remaining zero-dose prevalence, and identify risk factors limiting access and uptake. To address these gaps, a Post-Campaign Coverage Survey (PCCS) was conducted following Zambia's 2024 MR SIA. The survey was designed to provide an independent estimate of vaccination coverage among children aged 9–59 months, offering a validated measure beyond routine administrative reports. It also aimed to quantify the prevalence of zero-dose and under-immunised children, assess their overlap with RI performance, and evaluate the proportion reached during the campaign. In addition, the PCCS examined socio-demographic, geographic, and awareness-related factors associated with non-vaccination, generating insights into barriers to uptake and informing strategies to reduce missed opportunities for immunisation. By generating robust and nationally representative evidence, this study provides critical insights for

strengthening measles control in Zambia and contributes to the global evidence base on strategies to reduce zero-dose children in LMICs.

## Methods

### Study design

The 2024 MR PCCS was a cross-sectional nationally representative household survey to evaluate the September 2024 MR - SIA in Zambia. The survey followed the WHO Vaccination Coverage Cluster Survey Reference Manual [13] and was implemented between 27 December 2024 and 16 January 2025 across all 10 provinces. Zambia has an estimated population of 19,610,769 [14], administratively divided into provinces, districts, constituencies, and wards. Within wards, enumeration areas (EAs) mapped during the 2022 Census of Population and Housing served as the primary sampling units.

### Target population

The survey targeted children aged 9–59 months at the time of the September 2024 SIA. Eligible children were those born between 23 October 2019 and 23 December 2023. Household eligibility required at least one child under the age of seven years residing in the household.

### Sampling frame and sample design

The sampling frame was derived from the 2022 Census of Population and Housing. A two-stage stratified cluster sampling design was employed. In the first stage, 370 EAs (clusters) were selected using probability proportional to size (PPS), stratified by province and by urban/rural residence. In the second stage, systematic random sampling was used to select 25 households with at least one eligible child from each EA, after a complete household listing exercise. This yielded a planned sample of 9,250 households, providing adequate precision for national, provincial, and urban–rural level estimates.

### Sample weights

Sampling weights were calculated as the inverse of the product of the first-stage and second-stage selection probabilities. Adjustments were applied for non-response at the household level. All analyses incorporated sampling weights to generate population-representative estimates.

### Data collection tools

 A structured household questionnaire, adapted from the WHO PCCS standard tool, was programmed into the CSPro application for Computer-Assisted Personal Interviewing (CAPI). The instrument included modules on household demographics, caregiver education and occupation, child vaccination history (card verification or caregiver recall), knowledge and awareness of the SIA, sources of information, and barriers to vaccination uptake. Behavioural and social drivers of vaccination were assessed using items aligned with the WHO BeSD (Behavioural and Social Drivers of Vaccination) framework.

### Training and fieldwork

A total of 164 field staff (131 enumerators and 33 supervisors) underwent a six-day centralised training facilitated by the Zambia Statistics Agency and Ministry of Health. Training included survey objectives, questionnaire content, CAPI procedures, household listing, interviewing techniques, ethical considerations, and practical field exercises. Only participants who passed competency assessments were deployed. Field teams of 4–6 members conducted household listing and interviews under close supervision. Geographic Information Systems (GIS) staff verified EA coverage before household

selection. Data were collected electronically using tablets, with secure daily uploads to a central server at the Zambia Statistics Agency.

## Data quality assurance

Supervisors conducted daily field checks and verified completed interviews. Automated data consistency checks were embedded in the CAPI system. Data files were reviewed centrally for completeness, and errors were corrected before final cleaning. To minimise recall bias, enumerators were trained to use standardised probing techniques, verify vaccination status using any available home-based records, and capture photographs of cards to validate caregiver reports. Where cards were unavailable, interviewers applied event-anchoring techniques (e.g., timing relative to major local events or clinic visits) recommended in WHO PCCS guidelines to improve accuracy. As this was a cross-sectional coverage survey conducted after the campaign, some missing data occurred due to unavailable vaccination cards, incomplete caregiver recall, or occasional skipped items during interviews. These sources of missingness are typical in household immunisation surveys.

## Response rates

Of the 9,211 sampled households, 9,018 were successfully interviewed, yielding a household response rate of 97.8% (98.1% urban; 97.6% rural). A total of 8,634 eligible children were surveyed, representing 2.17 million children nationally when weighted.

## Statistical analysis

Survey-weighted analyses were conducted using the survey package in R (version 4.3.2). All estimates—proportions, odds ratios, and 95% confidence intervals—accounted for the complex survey design, including stratification by province and urban/rural residence, clustering at the enumeration-area level, and application of sampling weights. Variables considered for association with MR zero-dose status were selected based on prior literature and the WHO Behavioural and Social Drivers (BeSD) framework. These included:

i. demographic factors (child age [9–23, 24–36, > 36 months], sex, household size, maternal vital status);

ii. socioeconomic factors (guardian education, occupation-derived socioeconomic tertile, religion);

iii. geographic and access factors (province, urban/rural residence, travel time to the nearest vaccination post, usual mode of transport); and

iv. campaign awareness and barriers (exposure to SIA messaging, source of information, and reported reasons for non-vaccination).

Bivariable associations were assessed using survey-weighted logistic regression. Multivariable models were developed using a stepwise approach guided by both statistical criteria ($p < 0.10$ in bivariable analysis) and conceptual importance; for example, maternal status and travel time were retained regardless of bivariable p-values owing to strong theoretical relevance. Collinearity was assessed using variance inflation factors (VIF), and all included variables had VIF values below 3, indicating no concerning multicollinearity. Adjusted odds ratios with 95% confidence intervals are reported, with statistical significance set at $p < 0.05$. Missing data (below 3% for any covariate) were handled using listwise deletion; complete-case analysis was appropriate given the low level of missingness.

## Ethical considerations

The PCCS was implemented under the authority of the Zambia Ministry of Health as a public health surveillance activity and was not classified as human subjects research. Nevertheless, ethical principles were observed. Informed verbal

consent was obtained from all caregivers prior to interviews. All data were anonymised, and personal identifiers were not retained in analytic datasets.

## Results

### Zero-dose prevalence

Nationally, 11.97% (95% CI: 11.03–12.91%) of children were MR zero-dose, of whom 88.75% had not received Diphtheria, Pertussis and Tetanus first dose (DPT1) or other antigens, indicating systemic immunization gaps (Table 1, S2 Table). Prevalence was highest in Central (19.15%) and Western (17.71%) provinces, and lowest in Copperbelt (6.69%) (Fig 1). Rural prevalence (13.02%) exceeded urban (10.17%), with an adjusted odds ratio (OR) for urban residence of 0.76 (95% CI: 0.63–0.92), indicating higher odds of zero-dose status in rural settings (S1 Table). Children in large households (≥9 members) had a higher MR zero-dose prevalence (14.18%; 95% CI: 11.63–16.73%) compared to those in households with 1–4 members (12.29%; 95% CI: 10.72–13.86%) or 5–8 members (11.17%; 95% CI: 9.88–12.46%), though this association was not statistically significant (S1 Table). Stratified analysis showed similar patterns in both rural (≥9: 14.41%) and urban settings (≥9: 13.41%), confirming that the household size effect was consistent but not significant across residence types.

### Demographic risk factors

Measles-rubella (MR) zero-dose prevalence increased with age, peaking at 14.99% among children >36 months (OR = 1.60; 95% CI: 1.27–2.00) (Fig 2). No significant sex differences were observed between males (11.90%; 95% CI: 10.92–12.88%) and females (12.04%; 95% CI: 11.04–13.04%), with an adjusted odds ratio of 1.05 (95% CI: 0.89–1.23; p = 0.533) (S1 Table). Children whose mothers were absent (19.73%; OR = 1.74, 95% CI: 1.33–2.27) or deceased (24.17%; OR = 2.40, 95% CI: 1.23–4.68) had increased odds of being MR zero-dose (S1 Table). A small group with "unknown" maternal status showed zero vaccination (OR = 0.00), but this reflects unstable estimates due to very small sample size. Guardian education (e.g., Primary: 11.33%; 95% CI: 9.82–12.84%) and religion (e.g., Protestant: 12.50%; 95% CI: 11.44–13.56%) showed no statistically significant associations (S2 Table). Similarly, socioeconomic status (SES), derived

**Table 1. Key findings on MR zero-dose prevalence and adjusted associations, Zambia PCCS 2024 (n = 8,634).**

| Variable (comparison vs reference) | Zero-dose prevalence (%) | 95% CI (%) | Adjusted OR (95% CI) |
|---|---|---|---|
| National prevalence | 11.97 | 11.03–12.91 | — |
| Province – Central | 19.15 | 15.84–22.46 | — |
| Province – Copperbelt | 6.69 | 4.67–8.71 | — |
| Urban (vs Rural) | 10.17 | 9.29–11.05 | 0.76 (0.63–0.92) |
| Age > 36 months (vs 0–12 months) | 14.99 | 13.25–16.73 | 1.60 (1.27–2.00) |
| Mother absent (vs present) | 19.73 | 15.85–23.61 | 1.74 (1.33–2.27) |
| Mother deceased (vs present) | 24.17 | 12.60–35.74 | 2.40 (1.23–4.68) |
| Travel time >2 hours (vs < 15 minutes) | 25.71 | 10.94–40.48 | 3.20 (1.43–7.16) |
| Campaign awareness – Not heard (vs Heard) | 18.50 | 16.17–20.83 | 2.00 (1.65–2.42) |

1. Prevalence values are zero-dose prevalence within each subgroup (survey-weighted, with 95% CIs).

2. Adjusted ORs are from survey-weighted logistic regression accounting for stratification, clustering and sampling weights; reference categories shown in brackets.

3. Reasons for non-receipt (e.g., "Lack of awareness 42.58%") describe the distribution of reported reasons among zero-dose children and are not directly comparable to subgroup prevalence; report these in a separate table/figure.

4. Systemic gap marker: *Among zero-dose children, 88.75% had received no other routine vaccines.* If desired, add a one-line footnote: "Receipt of other routine vaccines (vs none) was associated with lower odds of MR zero-dose (aOR 0.26, 95% CI 0.08–0.91)."

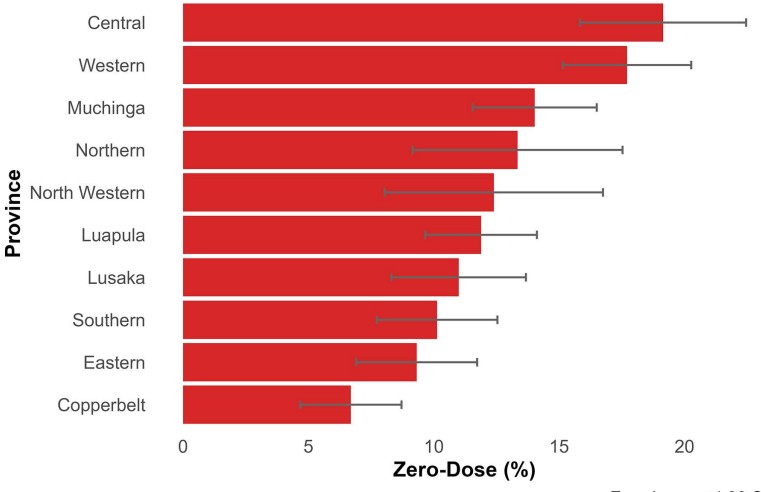

**Fig 1. Weighted zero-dose measles-rubella prevalence by province, Zambia, 2024 PCCS. Exact values are in S1 Table.**

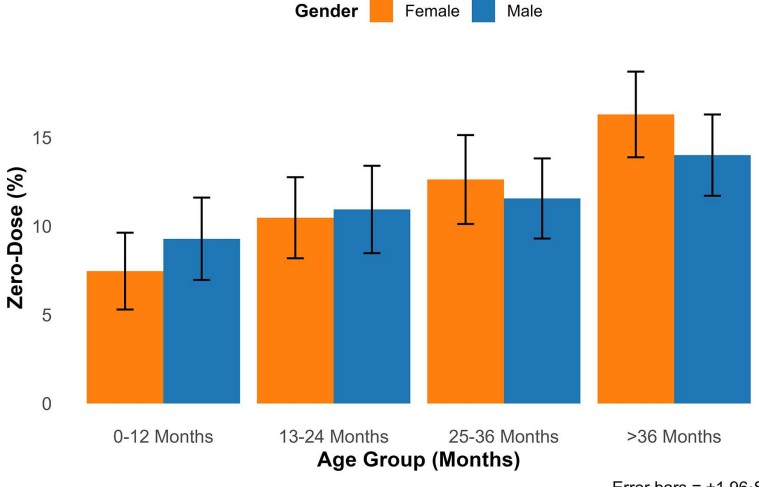

**Fig 2. Weighted percentage of MR zero-dose children by age and gender, Zambia, 2024 PCCS.**

from occupation, did not show a consistent pattern across low, medium, and high categories, and associations were not statistically significant. Muslim prevalence was unreliable (0.00%) due to small sample size (S2 Table).

## Systemic vaccination gaps

Among MR zero-dose children, 88.75% had not received any other basic vaccines, indicating broader immunisation failure rather than isolated MR campaign gaps. Those who received other vaccines had significantly lower odds of being MR zero-dose (OR = 0.26; 95% CI: 0.08–0.91).

## Barriers to vaccination

Lack of awareness of the campaign (42.58%) was the most cited reason for non-vaccination, followed by being too busy (19.02%) and inconvenient timing (7.69%) (Fig 3). Travel time of 1–2 hours (OR = 1.55; 95% CI: 1.20–2.00) and over 2 hours (OR = 3.20; 95% CI: 1.43–7.16) were associated with significantly higher odds of being MR zero-dose (S2 Table). The regression model also included travel means (mode of transport to the facility), but no significant associations were observed. CIs for >2 hours were wide due to small sample sizes.

## Discussion

Zambia's national MR zero-dose prevalence of 11.97%, with high-burden provinces such as Central (19.15%) and Western (17.71%), reflects persistent geographic inequities and structural barriers, including limited access, low campaign awareness, and social vulnerabilities like maternal absence, common in LMICs. This analysis highlights three critical findings: first, clustering of zero-dose children in specific provinces and among older children; second, the dominance of systemic access barriers such as distance and awareness; and third, the overlap of MR zero-dose status with complete lack of other vaccines (88.75%), signalling service delivery failures across the immunisation system rather than isolated campaign shortfalls.

In regional context, Zambia's 2024 burden (~41,000 zero-dose children) places it mid-range, higher than countries such as Botswana or Eswatini, but below large-burden nations like Ethiopia and Angola [5]. This comparative framing underscores the need for targeted but feasible interventions to reach IA2030 zero-dose reduction goals.

Older children and those with absent or deceased mothers were significantly more likely to be zero-dose, consistent with global evidence linking maternal absence/orphanhood to vaccination gaps [8]. These associations may reflect mechanisms such as reduced caregiver decision-making capacity, opportunity costs during agricultural seasons, or limited household support structures. By contrast, socioeconomic status and caregiver education were not significant, suggesting

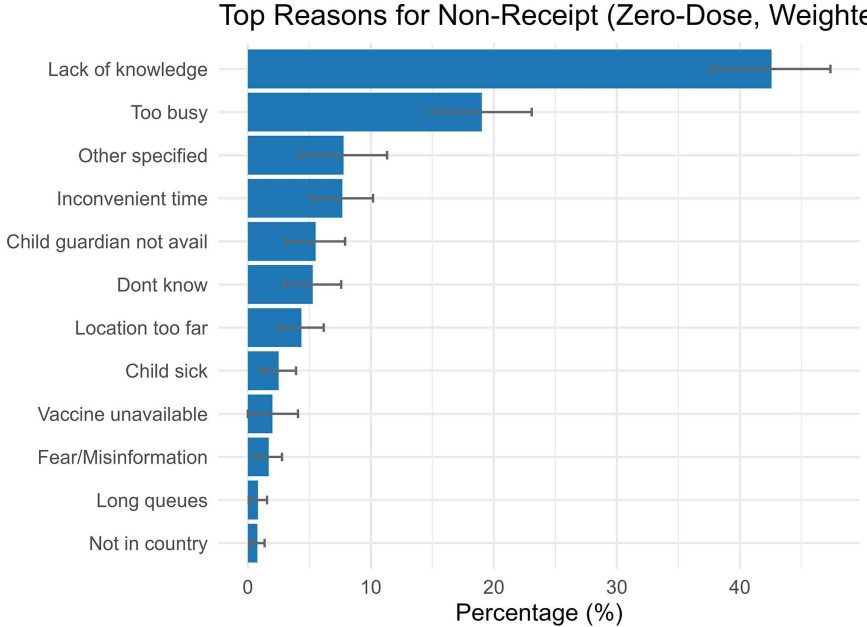

**Fig 3. Weighted reasons for non-receipt of measles-rubella vaccine among zero-dose children, Zambia, 2024 PCCS.**

that structural determinants such as distance, service timing, and awareness are stronger drivers than demand-side constraints in this setting.

The co-occurrence of zero-dose MR with absence of other antigens emphasises that Zambia's challenge lies in RI system failures. Weak microplanning, poor integration between SIAs and RI, and inconsistent card retention limit opportunities for catch-up. This aligns with global analyses indicating that SIAs often overestimate their independent contribution when RI remains weak [10,11].

Importantly, our interpretation of COVID-19–related disruptions requires clarification. Evidence shows that measles vaccination was strongly protective, with vaccinated children having markedly lower mortality (aOR ~0.06 compared to unvaccinated [15]) Zambia's pandemic-era gaps therefore exacerbated measles outcomes by reducing coverage, not by conferring protection to unvaccinated children.

Access barriers were central with children living >2 hours from a facility had a prevalence of 26% versus 10% within 15 minutes representing ~16 percentage-point difference (aOR 3.20). Awareness gaps also remain striking, with 42.6% of caregivers of zero-dose children reporting they had not heard of the campaign. Together, these patterns suggest that reducing travel time and improving health worker–led communication could yield greater gains than socioeconomic-targeted messaging alone. Similar patterns of access-related and caregiver-related determinants of under-vaccination have been documented in India, Ethiopia, and the Democratic Republic of the Congo, reinforcing the generalisability of these findings across diverse LMIC settings [16–21].

Generalisability is high for LMIC settings with dispersed rural populations, weak outreach systems, and heavy reliance on SIAs. In dense urban contexts with abundant fixed sites, demand-side determinants (e.g., confidence, complacency) may be more salient, suggesting that Zambia's lessons apply most strongly to rural, outreach-dependent systems.

This study makes three novel contributions: (i) it quantifies the overlap of MR zero-dose with all-antigen zero-dose nationally, (ii) it demonstrates the primacy of access and awareness barriers after adjusting for stratification, and (iii) it identifies older children and those without maternal caregivers as particularly high-risk groups requiring tailored catch-up strategies.

To close the remaining immunisation gaps, Zambia should prioritise strengthened, proactive communication led by health workers and tailored to local contexts, particularly in high-burden areas where awareness remains low, as supported by recent evidence showing that targeted health-worker–driven communication significantly improves uptake [22]. Furthermore, expanding proximity-based delivery strategies, such as regular mobile and outreach sessions, would reduce long travel times and improve accessibility, consistent with studies demonstrating the effectiveness of mobile teams in tracing unvaccinated children and increasing RI coverage [23]. In parallel, closer integration of Supplementary Immunisation Activities (SIAs) with routine immunisation (RI) is essential, ensuring that all SIA posts implement defaulter tracing and provide on-site catch-up for routine antigens, supported by improved card verification. Additionally, adopting geospatial microplanning approaches to generate and update hotspot maps quarterly would facilitate more precise targeting of older children and households without maternal caregivers, aligning with findings from recent geospatial analyses showing the value of high-resolution mapping for identifying zero-dose clusters [24,25]. Digital tools, including mobile applications for defaulter tracking and AI-supported hotspot identification, can also enhance microplanning efficiency, building on evidence from both Zambia and other LMICs that digital health tools increase coverage and strengthen planning [26], as well as Nigeria's documented success in reducing zero-dose gaps [27]. Moreover, adapting Zambia's cholera control coordination model to measles, leveraging community engagement and strengthened surveillance, could further improve RI performance in rural hotspots [28]. Collectively, embedding these equity-focused reforms within national immunisation strategies would enable SIAs to serve as more effective platforms for reinforcing RI, addressing systemic gaps, and accelerating progress toward measles elimination.

## Limitations

The study has several limitations. Although 88.3% of vaccination status assessments relied on caregiver recall, several mitigation strategies were used to reduce recall bias. Interviewers followed WHO-standardised probing procedures, captured

card images where available, and used event-anchoring to contextualise timing. These steps minimise but do not eliminate recall bias; therefore, the direction of potential misclassification is uncertain. However, the approach reflects real-world immunisation documentation challenges in Zambia. Small sample sizes in some subgroups, particularly for deceased mothers and certain provinces, limit the precision of stratified estimates. Additionally, district-level variations may be masked by provincial-level reporting, potentially obscuring important local patterns that could inform targeted interventions.

## Conclusion

This study set out to estimate the burden of measles–rubella zero-dose children following the 2024 supplementary immunisation activity and to identify the key factors associated with non-vaccination. The survey showed that a substantial group of children remained unreached, and these children were disproportionately concentrated in rural settings, specific high-burden provinces, older age groups, and households without maternal caregivers. The analysis also highlighted the influence of structural barriers, particularly limited awareness of the campaign and long travel distances to vaccination points, which undermined uptake despite the national effort.

Overall, the findings demonstrate that achieving measles control in Zambia requires strategies that strengthen RI while directly addressing the contextual barriers that limit access and awareness. Approaches centred on community engagement, improved communication, and enhanced service reach, especially for hard-to-access populations, will be essential for reducing zero-dose children and advancing progress toward national and global measles elimination goals.

## Supporting information

**S1 Table. Zero-dose measles–rubella prevalence by demographic and geographic characteristics, Zambia, 2024 PCCS.** Weighted MR zero-dose prevalence and 95% CIs by age, sex, province, maternal status, household size, and residence. Significant adjusted odds ratios from survey-weighted logistic regression are reported.
(DOCX)

**S2 Table. Zero-dose measles–rubella prevalence by socioeconomic and access characteristics, Zambia, 2024 PCCS.** Weighted MR zero-dose prevalence and 95% CIs by guardian education, campaign awareness, travel time, reasons for non-receipt, receipt of other antigens, and religion. Significant adjusted odds ratios are shown.
(DOCX)

**S1 Data. Summary of aggregated data used in analyses.**
(CSV)

## Acknowledgments

The authors extend their sincere gratitude to the staff of the Zambia Ministry of Health, WHO Zambia Country Office, UNICEF Zambia, and the Zambia Statistics Agency (ZAMSTATS) for their invaluable technical and logistical support throughout the 2024 Post-Campaign Coverage Survey. We also acknowledge the dedication of the data collectors and community health workers whose efforts made this study possible.

## Author contributions

**Conceptualization:** Moses Mwale, Princess Kayeye, Simon Mutembo, Freddie Masaninga.

**Data curation:** Moses Mwale.

**Formal analysis:** Moses Mwale.

**Funding acquisition:** Moses Mwale, Peter Clement Lugala.

**Investigation:** Moses Mwale, Guissimon Phiri, Penelope Masumbu, Kennedy Matanda, Princess Kayeye, Chola Nakazwe, Harriet Namukoko, Simon Mutembo.

**Methodology:** Moses Mwale, Penelope Masumbu, Kelvin Mwangilwa, Chola Nakazwe, Harriet Namukoko.

**Software:** Moses Mwale.

**Supervision:** Guissimon Phiri, Francis Dien Mwansa, Peter Jay Chipimo, Freddie Masaninga, Jacob Sakala, Peter Clement Lugala.

**Validation:** Penelope Masumbu.

**Writing – original draft:** Moses Mwale.

**Writing – review & editing:** Guissimon Phiri, Francis Dien Mwansa, Peter Jay Chipimo, Penelope Masumbu, Kennedy Matanda, Kelvin Mwangilwa, Harriet Namukoko, Simon Mutembo.

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
