## [Decision Letter · Decision Letter 0]

5 Nov 2025

PGPH-D-25-02721

Profiling Zero-Dose Measles-Rubella Children in Zambia: Insights from the 2024 Post-Campaign Coverage Survey

Dear Dr. Mwale,

Thank you for submitting your manuscript to PLOS Global Public Health. After careful consideration, we feel that it has merit but does not fully meet PLOS Global Public Health’s publication criteria as it currently stands. Therefore, we invite you to submit a revised version of the manuscript that addresses the points raised during the review process.

The reviewers have relatively minor comments - but I agree that the discussion should be put in more narrative form - and try to include a few more references for your recommendations.

We look forward to receiving your revised manuscript.

Kind regards,

Abram L. Wagner, PhD, MPH

Academic Editor

Journal Requirements:

1. In the online submission form, you indicated that “All data underlying the findings are available upon request”.

a) In a public repository,

b) Within the manuscript itself, or

c) Uploaded as supplementary information.

2. Please provide separate main figure files in .tif or .eps format only and ensure that all files are under our size limit of 10MB.

3. Please ensure that the Author List in your manuscript file matches the Author List in the online submission form exactly. Authors should be listed in the same order, and the order of individual first and last author names must be identical in both locations. 

4. We notice that your supplementary tables are included in the manuscript file. Please remove them and upload them with the file type 'Supporting Information'. Please ensure that each Supporting Information file has a legend listed in the manuscript before or after the references list.

Additional Editor Comments (if provided):

Reviewers' comments:

Reviewer's Responses to Questions

**Comments to the Author**

1. Does this manuscript meet PLOS Global Public Health’s publication criteria?

Reviewer #1: Yes

Reviewer #2: Yes

Reviewer #3: Yes

2. Has the statistical analysis been performed appropriately and rigorously?

Reviewer #1: I don't know

Reviewer #2: Yes

Reviewer #3: Yes

3. Have the authors made all data underlying the findings in their manuscript fully available (please refer to the Data Availability Statement at the start of the manuscript PDF file)?

Reviewer #1: Yes

Reviewer #2: Yes

Reviewer #3: Yes

4. Is the manuscript presented in an intelligible fashion and written in standard English?

Reviewer #1: Yes

Reviewer #2: Yes

Reviewer #3: Yes

Reviewer #1: The article “Profiling Zero-Dose Measles-Rubella Children in Zambia: Insights from the 2024 Post-Campaign Coverage Survey” addresses an urgent global health issue aligned with IA2030 and Gavi’s zero-dose priorities. The title is concise, descriptive, and fits the scope of PLOS Global Public Health (PGPH).

The study employs a cross-sectional, two-stage cluster survey following WHO guidelines, with robust sample size (n=8,634) and weighting for representativeness. Statistical analyses—survey-weighted logistic regression and confidence intervals—are appropriate. Ethical standards and data quality controls are well-documented. However, heavy reliance on caregiver recall (88.3%) introduces recall bias, and the absence of district-level disaggregation limits local applicability. The manuscript’s use of WHO standards and analytical transparency strengthens credibility.

It provides novel national evidence on MR zero-dose prevalence and systemic immunization failures in Zambia, filling a gap between administrative and survey estimates. The identification of access and awareness barriers (e.g., 42.6% unaware of campaigns) adds actionable insights for health

The article follows a clear IMRaD structure with strong coherence between results, discussion, and policy recommendations. Figures and tables are informative, though data visualization could be simplified for readability. Language is clear and professional, though some sections (e.g., policy implications) could be condensed to reduce redundancy.

No conflicts of interest or funding bias reported. Data availability upon request aligns with journal policy, though full open-access data would enhance transparency.

Overall, the manuscript is methodologically sound, policy-relevant, and well-aligned with PGPH’s thematic focus on equity and immunization coverage. Minor revisions are recommended—clarify recall bias mitigation, improve data visualization, and emphasize data accessibility. With these revisions, it is highly suitable for publication in PLOS Global Public Health.

Reviewer #2: 1. Is this a national wide surgvey? Please explain.

2. Does this survey cover whole population? or part of population? What is the percentage of coverage?

3. The survey is procpective study. How does it happen to miss the data? Please expalin.

4. The ststistical part need more elaboration considering the variables.

5. In discussion section, avoid bullet. Avoid the policy implecation rather right as paragraph.

6. Rewrite the conclusion. Avoid frequency, percent, only mentiont he fidnings in relation to the objective of the study.

Reviewer #3: In the current era of changing global and public health landscape, this manuscript is very timely in helping Zambia to improve vaccination coverage and address the inequities that exacerbates children to miss vaccinations. The manuscript is nearly perfect for publication with exception of few editorial areas which I request the authors to work on before the manuscript gets published. The areas are highlighted below:

ABSTRACT

Background

Line 22: I suggest to add the abbreviation MR in brackets after “Rubella”.

Line 23: I suggest to replace “and” with “can” between communities & sustain.

Methods

Line 27: suggest to insert “was conducted from” before 27th & replace “-” with “to” between 2024 and 16th.

Conclusions

Line 42: I suggest to write “RI” in its long form and the abbreviation in brackets.

Line 44: I suggest to edit “IA2030” to be “Immunization Agenda 2030”.

INTRODUCTION

Line 86: I suggest to insert the abbreviation “RI” in brackets after “immunisation” before “performance”.

METHODS

Study Design

Line 97: I suggest to replace “Post-Campaign Coverage Survey (PCCS)” with its abbreviation PCCS.

Lines 98-99: I suggest to replace “Measles–Rubella (MR) Supplementary Immunisation Activity (SIA)” with the abbreviations “MR – SIA.

RESULTS

Zero-Dose Prevalence

Line 172: I suggest to write “DPT” in its long form and the abbreviation in brackets.

DISCUSSION

Line 262: I suggest to replace “routine immunisation” with the abbreviation “RI”.

CONCLUSION

Line 325: I suggest to replace “routine immunisation” with the abbreviation “RI”.

**Do you want your identity to be public for this peer review?** For information about this choice, including consent withdrawal, please see our Privacy Policy

Reviewer #1: No

Reviewer #2: **Yes: ** Md Abdullah Yusuf

Reviewer #3: **Yes: ** Eliudi Saria Eliakimu

---

## [Editor Report · Decision Letter 1]

3 Dec 2025

Profiling Zero-Dose Measles-Rubella Children in Zambia: Insights from the 2024 Post-Campaign Coverage Survey

PGPH-D-25-02721R1

Dear Mr Mwale,

We are pleased to inform you that your manuscript 'Profiling Zero-Dose Measles-Rubella Children in Zambia: Insights from the 2024 Post-Campaign Coverage Survey' has been provisionally accepted for publication in PLOS Global Public Health.

Best regards,

Abram L. Wagner, PhD, MPH

Academic Editor